# Management of Non-Melanoma Skin Cancer: Radiologists Challenging and Risk Assessment

**DOI:** 10.3390/diagnostics13040793

**Published:** 2023-02-20

**Authors:** Gaetano Maria Russo, Anna Russo, Fabrizio Urraro, Fabrizio Cioce, Luigi Gallo, Maria Paola Belfiore, Angelo Sangiovanni, Stefania Napolitano, Teresa Troiani, Pasquale Verolino, Antonello Sica, Gabriella Brancaccio, Giulia Briatico, Valerio Nardone, Alfonso Reginelli

**Affiliations:** 1Department of Precision Medicine, University of Campania “Luigi Vanvitelli”, 80123 Naples, Italy; 2Unit of Plastic Surgery, Multidisciplinary Department of Medical Surgical and Dental Specialties, University of Campania “Luigi Vanvitelli”, 80120 Naples, Italy; 3Unit of Dermatology, University of Campania “Luigi Vanvitelli”, 80123 Naples, Italy

**Keywords:** dermatology, high-frequency ultrasound, MDT, skin cancer, radiotherapy, melanoma, oncology, ultrasound, computed tomography, magnetic resonance imaging

## Abstract

Basal cell carcinoma, squamous cell carcinoma, and Merkel cell carcinoma are the three main types of nonmelanoma skin cancers and their rates of occurrence and mortality have been steadily rising over the past few decades. For radiologists, it is still difficult to treat patients with advanced nonmelanoma skin cancer. Nonmelanoma skin cancer patients would benefit greatly from an improved diagnostic imaging-based risk stratification and staging method that takes into account patient characteristics. The risk is especially elevated among those who previously received systemic treatment or phototherapy. Systemic treatments, including biologic therapies and methotrexate (MTX), are effective in managing immune-mediated diseases; however, they may increase susceptibility to NMSC due to immunosuppression or other factors. Risk stratification and staging tools are crucial in treatment planning and prognostic evaluation. PET/CT appears more sensitive and superior to CT and MRI for nodal and distant metastasis as well as in surveillance after surgery. The patient treatment response improved with advent and utilization of immunotherapy and different immune-specific criteria are established to standardized evaluation criteria of clinical trials but none of them have been utilized routinely with immunotherapy. The advent of immunotherapy has also arisen new critical issues for radiologists, such as atypical response pattern, pseudo-progression, as well as immune-related adverse events that require early identification to optimize and improve patient prognosis and management. It is important for radiologists to have knowledge of the radiologic features site of the tumor, clinical stage, histological subtype, and any high-risk features to assess immunotherapy treatment response and immune-related adverse events.

## 1. Introduction

Cancer ranks as a major cause of mortality worldwide and consequently an important barrier to growing life expectancy in every country [1]. Estimated data from the World Health Organization (WHO) in 2019 represent cancer as the first or second major cause of death in 112 out of 183 countries before the age of 70s [2]. Skin carcinomas, despite prevention and early detection, are common malignant tumors globally and incidence rates have been increasing in recent decades [3]. Skin carcinomas are differentiated into two types: melanoma skin cancer (MSC) and nonmelanoma skin cancer (NMSC) [4]. The incidence rate of nonmelanoma is 18–20 times higher compared to melanoma skin cancer [5].

The Global Cancer Statistics 2020 study by Sung et al. [1] reported that the prevalence of nonmelanoma skin cancer (NMSC) continues to increase in past years and is responsible for over one million new cases and 64,000 mortalities every year in the world. The NMSC includes basal cell cancers, squamous cell cancers, Merkel cell carcinomas, malignant adnexal tumors, and other rarer skin neoplasms. However, basal cell cancers (BCC) and squamous cell cancers (SCC) are the most prevalent types of carcinomas which account for 99% of all NMSCs [6].

These carcinomas originate from the skin epidermal cells and have common epidemiological and carcinogenic features as well as have a worse prognosis. Besides having a high incidence rate, most of the large cancer registries lack NMSC epidemiological data or exclude NMSC from their records due to significant limitations in establishing the large number of cases [6,7]. The risk factors associated with increased NMSC incident rate include high ultraviolet (UV) or sunlight exposure [8], high outdoor activities, ozone depletion [9], genetic mutation, and immune suppression, as well as correlated with other various factors such as dose of UV radiation, age, skin phenotype, degree, and chronicity [10,11].

Genomic defects identified either in the germline or somatic mutation are predisposing causes of developing nonmelanoma skin cancer [12]. Tumor size, depth, and thickness, as well as anatomic involvement with lymphatic circulation, are predictive in risk stratification, TNM staging, evaluation of prognosis, and recurrence rate of NMSC after treatment [12,13,14].

NMSC, locally invasive carcinomas, is growing slowly and ulcerates in body parts. Early detection improves prognosis by having a range of treatment choices and thus reduces morbidity and health costs and improves the quality of life. Although metastasis in NMSC is rare, diagnosis at advanced stages with the involvement of other anatomical parts makes management of NMSC difficult and worsens the prognosis [15].

High-risk NMSC may invade local anatomic structures, more, about 5% of squamous cell carcinomas have a high risk of distant metastasis [16,17,18,19] and indicate optimal diagnostic imaging, including computed tomography and magnetic resonance imaging. Low-risk NMSCs at the earliest stage can be managed without diagnostic imaging [19].

Imaging is required when the risk of invasion is concerned. Imaging is an indispensable tool for the detection of nodal and distant disease and staging of aggressive neoplasms. Thus, to avoid unnecessary invasive treatment, improving pre-surgical accuracy of lesion characterization and management of patients with NMSC demands early diagnostic imaging [20] and it is clear that diagnostic imaging has a pivotal role in management of disease. Though, with the introduction of a multidisciplinary patient management approach, the role of the radiologist is profoundly changed in patient-focused care and becomes part of patient management [21,22,23,24,25,26,27,28].

Radiologists have a critical role in detection, screening, staging, and management of cancer patients with a wide range of radiology modalities or tools, utilized for initial screening, follow-up by staging, and surveillance of the extent of disease, moreover, in selection of the cancer treatment regimen, pre-treatment planning consistent with the tumor-node-metastasis (TNM) system in the oncology patient.

However, some critical issues concerning the radiologist’s role in nonmelanoma skin cancer management should be analyzed. Firstly, the radiologist’s role in patient management, presenting different phases of NMSC, based on staging, surveillance, or follow-up, as well as assessment of treatment effectiveness and observation of treatment-related adverse events or complications [29,30,31,32,33].

Secondly, the application of a different diagnostic tool should be used for detection and risk assessment [34,35,36,37,38,39,40]. NMSCs assessment time depends on the patient risk and treatment type employed. The aim of this review paper is a critical analysis of the radiologist’s role in nonmelanoma skin cancer (NMSC) patients related to the different patient risk and disease phases.

## 2. Diagnostic Imaging and Non-Melanoma Skin Cancer

### Staging and Risk Stratification

Lesion characterization, differentiation, and risk stratification are required for future clinical decision-making, surgical versus nonsurgical lesion management, and for prognostic evaluation, more, it is employed as a reporting tool in institutional, national, and international cancer registries that aid in understanding NMSC epidemiology. The staging and risk stratification of NMSCs is based on their clinical-pathological features that are defined by National Comprehensive Cancer Network (NCCN) guidelines (2014) to differentiate low and high-risk carcinomas recurrence and metastasis [41,42,43].

NCCN guidelines are significant for basal cell carcinoma (BCC) risk stratification, management, and prognostic information as it often requires staging due to less incidence of metastasis [42,43]. However, squamous cell carcinomas are malignant and have the potential for distant metastasis, so the American Joint Committee on Cancers (AJCC) Cancer Staging Manual 8th edition published in 2017 revised the tumor, nodal, and metastases (TNM) staging of SCC concerning high-risk clinicopathologic features [19,44,45,46]. A whole-body skin physical examination or care inspection is required to evaluate and assess NMSC, suspicious lesions, satellites, regional lymph node (LN) in transit (ITM) and systemic metastases.

Tumor margins and depth of invasion were evaluated based on diagnostic imaging, but “T” category staging assessment relies on imaging assessment besides recent imaging advancements in technology [47,48,49,50] (Figure 1).

Many NMSCs could be managed without additional information provided by imaging and are not required in low-risk patients (pT1a). However, in pT1b to pT4b stages, additional information is necessary to optimize management, so ultrasound, computed tomography (CT) scan, positron emission tomography (PET) studies, as well as magnetic resonance imaging (MRI) are optional imaging modalities before operation or sentinel node biopsy (SNB). In a very high-risk patient with other anatomic parts involvement, brain MRI and PET-CT/CT are anticipated modalities [51,52].

## 3. Diagnostic Tools and Non-Melanoma: Staging and Surveillance

High-frequency US (HFUS) utilizing 20 to 100 MHz frequencies [53] could be a powerful tool for an accurate evaluation of the tumoral margins, diameters, and thickness, and provides information about deeper structures involvement as well as improves the performance of loco-regional staging [51,54,55] (Figure 2).

Accurate preoperative tumor assessment minimizes the extent of surgical defects and improves the cosmetic prognosis of patients. A retrospective study by Zhu et al. [56] analyzed HFUS features of actinic keratosis (AK), SCC in situ, and SCC and suggested that HFUS has good diagnostic accuracy with 85.3–92.3% sensitivity and 73.6–88.0% specificity based on diagnostic features.

The application of high-frequency US (HFUS) in nonmelanoma skin cancer facilitated high diagnostic accuracy in NMSC phase staging [57,58,59,60,61]. However, biopsy and histopathological analysis are gold standards and compulsive for correct staging even if HFUS detects nodular features to confirm the diagnosis [58,62]. As it lacks functional contrast and has low image resolution and quality, in addition, HFUS is highly operator-dependent and requires high expertise [59,63].

A comparative study by Crisan et al. [64] evaluated 18 BCC lesions depth index using HFUS, correlated ultrasonographic index with histological index, they reported a strong correlation between them. However, values obtained by using HFUS for tumor depth are less than values obtained by histological analysis. Another retrospective study by Wortsman [65] and his colleague reported the limited diagnostic ability of HFUS for the detection of malignant lesions.

So, HFUS may not measure the appropriate tumor depth precisely identical to histological examination and overestimating lesion depth or thickness as high probe compression shallows the lesion, more, the inflammatory response near the lesion is indistinguishable from tumor invasion by HFUS [64,66]. Moreover, HFUS demands high expertise with good SCC, BCC, and actinic keratosis sonographic features knowledge to recognize the normal structures as well as regional lymph nodes draining the skin [53]. In NMSC, lymph node involvement is a poor prognostic factor and it is suggested that nodal metastasis (N) mostly present in SCC and Merkel cell carcinomas are stronger predictors of prognosis than features present in the primary tumor [67]. Although the sentinel node is the first lymph node draining the tumor, the role of Sentinel lymph node biopsy and lymphatic imaging is not clearas it is in melanoma [68].

A retrospective study by Foy et al. [69] including 33 patients with high-risk head and neck nonmelanoma skin cancer patients suggested that SLNB is relevant in the management of N0 high-risk NMSC of the head and neck and should be performed in certain cases which are clinically suspicious or in patients with high-risk pathologic findings, such as lymphatic invasion. 

Early metastasis in the sentinel lymph node (SNL) demonstrates an elongated tumor due to immune cell aggregation [70], with a low cross-sectional area, and remains undetectable by current diagnostic tools. Such lesions could be evaluated indirectly by doppler as the metastatic lesions have increased vascularity and show an enhanced vascular signal [54]. However, doppler ultrasound is not helpful in differentiation of benign lesions from malignant ones. Ultrasound, in the detection and staging of NMSCs, is more sensitive and specific than physical evaluation [71,72,73,74] and is considered superior in the detection of lymph nodes metastasis during NMSC surveillance compared to other diagnostic tools such as CT or PET/CT scan [75,76] (Figure 3).

The diagnostic imaging follow-up, surveillance, and staging goal are usually correlated to lesion detections, staging phase, treatment response, and relapse rate provided. The values of all the above are well understood and evident, but all of them should be considered, concerning patient risk to ionizing radiations or imaging scan costs with different imaging phases. Compared to ultrasound, imaging scan costs of CT, MRI, and PET/CT are twofold, although no appropriate data on clinical outcomes exist to excuse higher cost modalities over patient health; and MRI, CT, and PET/CT are widely utilized modalities in patient management with metastatic disease for diagnostic and assessing treatment response [77,78].

Ruiz et al. [79] suggested the increase in the 5-year cancer-free survival rate in 78% patients who underwent imaging compared to 51% of patients who did not undergo imaging and reported alter in treatment plan of 33% of high-risk SCC patients with T2b/T3 staging phase, underwent diagnostic imaging scan. Therefore, the study demonstrates earlier detection, risk stratification, and treatment of advanced conditions, which have positive outcomes in patient tumor management. CT (79%), PET/CT (21%), and MRI (19%) were imaging modalities that were frequently used for diagnostic imaging.

In NMSCs, diagnostic imaging is performed in high-risk patients with suspicion of bony or other soft invasions, such as suspicious perineural invasion, therefore, an MRI (Figure 4, Figure 5 and Figure 6) scan is performed to assess the lesion extension and depth in soft tissue; and positron emission tomography (PET)/CT, 18F-fluorodeoxyglucose (FDG) is helpful in assessing nodal and distant metastases [78,80]. CT examination is preferred only to evaluate bone invasion due to its poor soft tissue contrast, which makes it difficult to evaluate NMSCs lesions and demonstrate non-specific tissue density [81].

Compared to CT scans, MRI provides good soft tissue contrast and is a good helping tool for examination or evaluation of configuration, intra-tumoral homogeneity, signal intensity, cyst formation, and haemorrhage [82]. The study by Rajesh et al. [83] demonstrated significantly shorter survival among patients with clinically suspected perineural invasion with squamous cell carcinoma and basal cell carcinomas, with imaging evidence of perineural spread.

Flat elevated lesions, multiple skin lesions, superficial depressions, and pedunculated configuration in SCC, MCC, and adnexal tumous, such as porocarcinoma, are observed in MRI [84,85]. BCCs and SCCs represent heterogeneously hyperintense signal intensity on the T2-weighted (T2W) scan, whereas hypointense signal intensity onT1-weighted (T1W) compared with muscles after the contrast agent injection in the MRI scan [86] (Figure 7). However, MCCs show homogeneous lesions with slight hyperintense signal intensity on the T1-weighted (T1W) scan and hyperintense signal intensity on T2-weighted images compared to muscles after administration of gadolinium contrast agent. Whereas larger lesions resulting in necrosis appear as a heterogeneous enhancement [87].

When carcinoma invades muscles and critical body structures, cross-sectional MRI imaging has a critical role in estimating the extent of disease margins and depth as well as provides quite accurate clinical information that is helpful regarding type and extent of surgery required [88,89]. MRI is good but expensive compared to other tools and takes a longer scan time. In addition, there are only a few research studies that demonstrated MRI features of SCC, BCC, and MCC [86,87], lack significant data to demonstrate MRI features of skin carcinomas that limit the use of MRI in this clinical setting and is challenging for radiologists in patient management with NMSCs.

PET (FDG) adjunct with CT provides functional and structural information and proves to be a helpful tool in evaluating nodal and distant metastases and unmasking occult recurrences or micro-metastasis [90]. NMSCs have a high metabolic rate and show high FDG uptake on PET/CT scan, with an average standardized uptake value (SUV) (7.6) in advanced stages.

PET/CT scan is advised clinically in patients with NMSC to diagnose distant metastasis, has high sensitivity in assessing the primary and recurrent squamous cell carcinomas with nodal disease, and plays a critical role in lesion management in about 22–28% of cases [91,92]. BCCs have low metastasis potential, so PET/CT is not recommended for BCCs. The maximum standardized uptake value (SUV) of recurrence SCC (6.4) is less than primary SCC without recurrence (13.0) [80], whereas SUV for MCCs ranges from about 4 to 6.5 [85]. A systematic review by Schröer-Günther et al. [93] including 1155 patients reported PET and PET/CT sensitivity and specificity ranging from 68 to 87% and 92 to 98%, respectively, in patients with stage III and IV malignant melanoma. However, such systematic review data are lacking for nonmelanoma skin cancer in high-risk patients to demonstrate PET and PET/CT diagnostic accuracy, sensitivity, and specificity. However, the sensitivity of FDG-PET is based on the size of the tumoral lesion, its anatomical location, and rate of FDG standardized uptake value. In short, PET/CT is superior to MRI and CT in evaluation of nodal metastasis, regional nodes, and distant metastasis but is limited in low-risk carcinomas and early stages of carcinomas (Figure 8).

## 4. Follow-Up and Surveillance: Time

At present, in patients after treatment with nonmelanoma skin cancer, 6- to 12-month intervals of clinical follow-up are recommended according to NCCN guidelines for detecting recurrent carcinoma and new lesions [94]. However, there is no clear evidence of time and diagnostic tool application in the follow-up. The surveillance proposal differs regarding risk assessment of patients after treatment as it does during the first 3 years, every 3 months visit, and 6–12 months in thereafter [95] because the probability of recurrence for SCC is 95% within 5 years, with 70% of recurrence within the first 2 years. In addition, the recurrences rate for BCC is greater than 5 years and requires a long-term follow-up [96]. A meta-analysis by McCusker et al. [97] including 100 patients with metastatic BCC reported an average 54-month survival period, which greatly varies among patients with regional metastasis (87 months) and distant metastasis with a 24-month survival period.

Routine imaging surveillance is not recommended in lower-risk patients with small-sized (thin) lesions. However, in high-risk patients, diagnostic imaging such as ultrasound, CT, or PET/CT scan is performed for early detection of recurrence and metastasis to improve the prognosis and patient survival rate [51,52]. Patients with aggressive head and neck squamous cell carcinomas required close follow-up for early evaluation of recurrent disease [98]. Postoperatively scarring, fibrosis, and altered local anatomy make it difficult to detect recurrences in CT and MRI. Thus, PET/CT facilitates and remains useful in the early detection of recurrent head and neck lesions, local skin recurrence, and distant spread after surgery [99,100]. Shintani et al. [101] studied the utility of early after surgical resection and demonstrated that nodes detected in PET/CT were histologically proven positive in 46% and early scans after surgery changes the treatment plan and management in 15% of patients. However, the efficiency of PET/CT is altered by tumor histology [16].

## 5. Treatment Assessment of NMSCs in Immunotherapy

Currently, a multidisciplinary approach is applied for the treatment and management of NMSC, including surgical excision, photodynamic therapy, chemotherapy, and radiotherapy [102]. BCC and SCC are frequently treated with curative surgery and radiotherapy and usually appear as localized tumors but MCC is a rare aggressive NMSC, present with nodal and distant metastasis at advanced stage [102,103].

The treatment of NMSCs is planned according to the patient’s stage of disease, but a patient with advanced stage disease has a relatively poor prognosis. Furthermore, patients with locally advanced lesions are not eligible for surgery or radiotherapy, which highlights the need for new treatment technologies. Different clinical trials demonstrate immunotherapy and targeted therapy as promising treatments for patients with locally advanced unresectable NMSCs [104,105,106,107,108,109,110].

The goal of immunotherapy in oncology patients is stimulation of the immune system involving complex multiple processes, including the utilization of immune checkpoint blockades (ICB), that permit stronger immune responses and these inhibitors activate the body’s own T-cells to attack carcinomas [111]. Consequently, a number of immune cells infiltrate at the tumor sites followed by tumor size reductions, causing an atypical response pattern in imaging studies, termed pseudo-progression in which an increase in lesion size or new tumor growth is observed [112,113].

Pseudo-progression is an atypical immune response observed in 10% of oncology patients receiving immunotherapy [114,115,116,117] and is challenging for radiologists to accurately detect due to the lack of biomarkers to differentiate the pseudo-progression and hyper-progression [112]. Hyper-progression, opposite to pseudo-progression, means tumor burden or growth is increased 2-fold after immunotherapy and occurs in severe true progression of lesion burdens [112] (Figure 9, Figure 10, Figure 11 and Figure 12; Table 1).

The treatment response criteria used to assess the chemotherapy effectiveness in overall reduction of tumor burden, lesion size, appearance of new lesions as disease progression are included in World Health Organization (WHO) criteria [118,119,120,121] and Response Evaluation Criteria in Solid Tumors (RECIST) [122,123]. However, it is notable that these treatment response criteria guidelines have limitations in patients treated with immunotherapy, as RECIST does not explain a treatment failure due to hyper-progression [121]. Consequently, RECIST criteria have been modified with specific immune-related response criteria (irRC) established, including immune-modified RECIST (imRECIST), to evaluate immunotherapy treatment response [124,125].

Specific immune-related response criteria (irRC) based on the WHO immune-specific criteria, featuring the possibility to continue the treatment after first radiologic progression documented with 5 mm least lesion size and need confirmatory imaging at least four weeks after progression evidence, furthermore, new tumor appearance is incorporated into the sum of total tumor burden and is not taken as disease progression [123,125]. However, increased estimation inconsistency occurs due to bi-dimensional assessment of target lesions in regards to unidimensional assessment of RECIST, and is the main limitation of these new criteria, as well these criteria could render difficulties in comparison in the majority of immunotherapy trials utilizing RECIST, and in progress at that time [124].

Therefore, immune-related RECIST (irRECIST), with a new feature as an increase of 20% in the total measurable tumor burden from nadir with a minimum of 5 mm and new lesion appearance, has been started as an immune-related Progression Disease (irPD) [126]. As per irRC and irPD criteria, disease progression (DP) is suggested after 4 weeks after imaging of the first evident progression, demonstrating new unequivocal progression (UEP) compared to prior assessment or another new lesion appearance [126].

A consensus guideline was promoted by the RECIST working group along with immunotherapy subcommittees to standardize data reading between different trials assessing immunotherapy and established a modification of RECIST1.1 (2009), immune RECIST (iRECIST) [123]. The iRECIST is the standard guideline criteria for immunotherapy trials including immune complete response (iCR), immune-stable Disease (iSD), immune-progression rate (iPR), and unconfirmed progression disease (iUPD) or confirmed progression disease (iCPD). The main advantages iRECIST provide are unconfirmed progression disease should be confirmed with a repeated imaging assessment for least 4–8 weeks from unconfirmed progression disease several times until confirmed progression disease by follow up scan [123]. With regard to new lesions that were detected and stated as iUPD, if the patient is stable, asymptomatic, and continues on immunotherapy. Later on, iUPD follow-up, if another new lesion is detected, targeted or non-targeted lesion size increase than 5 mm, the patient will be categorized in iCPD [124] (Table 2).

FDG PET/CT is often performed at baseline to evaluate overall tumor burden after 2–3 cycles in an immunotherapy response assessment [111], based on the change in size and FDG uptake (SUV), we could evaluate immunotherapy treatment response and residual metabolic activity. Mostly, immune-related criteria lack prospective justification as they are based on small cohorts. Although, the above immune-related criteria (ir) are standardized but useful only for evaluation between clinical trials with immunotherapy, and are not utilized or applicable in a routine treatment response assessment. Thus, risk stratification based on immunotherapy response requires future studies to validate treatment response assessment criteria [127].

## 6. Imaging of Immune—Related Adverse Events

Although immunotherapy improved NMSCs management outcomes, however, such treatments have also initiated an immune related adverse event (Table 3), by reactivating the immune system and development of toxicities profiles, involving different organ systems from head to toe [128,129], but the skin–lung gastrointestinal tract are more prone to these events.

Common Terminology Criteria for Adverse Events (CTCAE v5.0) are utilized to categorized ir-AEs as well as permits to compare toxicities in clinical trials related to immunotherapy [130]. A systematic review by Arnaud-Coffin et al. [131] reported the ir-AEs developed in patients treated with anti-PD-(L)1 inhibitors, anti-CTLA-4 inhibitors, combination immune checkpoints inhibition at a rate of 74%, 89%, and 90%, respectively. Different organs are prone to different types of immunotherapies, such as patients treated with anti–PD-1/PDL-1 therapy who may have pneumonitis and thyroid disorders, whereas patients treated with anti-CTLA-4 antibodies may have hypophysitis and colitis as ir-AEs [132]. Such adverse events could be detected prior to onset of symptoms by diagnostic imaging and early detection permits optimal management [133]. Therefore, radiologists must be aware of radiological features of such adverse events to evidence properly. X-ray may be a baseline tool for evaluation of these ir-AEs but theCT scan is more sensitive in detection of these events and differentiating different subtypes [134].

The ir-AEs knowledge is still in the embryonic stage and in the clinical setting, the process of ir-AEs is considered due to induction of autoimmune [135] and thus increasing the demand of radiologists to have background information related to imaging features of ir-AEs for appropriate management of patients within a multidisciplinary approach [136].

Future research studies are required to increase knowledge related to ir-AEs in healthcare professionals, including radiologists, in addition to understanding specific toxicities during immunotherapy, molecular mechanisms of ir-AEs, recognizing risk factors to improve safety profiles, developing an appropriate diagnostic tool, and defining optimal management and monitoring strategies for specific types of irAEs.

## 7. Conclusions

The management of patients with nonmelanoma skin cancer in an advanced stage of disease remains challenging for radiologists. It is important to develop optimized risk stratification and a staging tool with diagnostic imaging for nonmelanoma skin cancer patients in relation to the type of patient, and enhanced knowledge. The patient treatment response improved with utilization of immunotherapy, however, new critical issues have arisen for radiologists, such as the atypical response pattern, pseudo-progression, as well as immune-related adverse events that require early identification for optimized and improved patient prognosis and management.

## Figures and Tables

**Figure 1 diagnostics-13-00793-f001:**
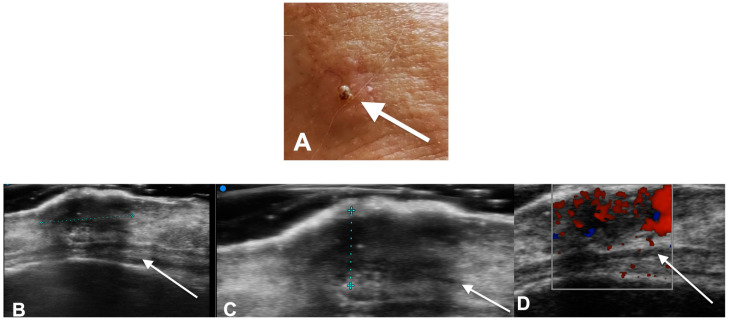
(**A**) A 73-year-old male patient. Right frontal diffuse infiltrative squamous cell carcinoma 444 two slightly hyperchromic hard and hypomobile areas to the underlying floors. (**B**–**D**) HF Ultra-445 sound examination performed with a very high frequency probe (48 Mhz). (**B**) Right upper frontal 446 site hypoechoic area of the hypodermis with blurred margins lower. (**C**) Non-encapsulated of the 447 following dimensions: Dt 6.9 mm × DL2.4 mm. (**D**) The lesion shows intralesional micro-vasculature 448 on Color Doppler control.

**Figure 2 diagnostics-13-00793-f002:**
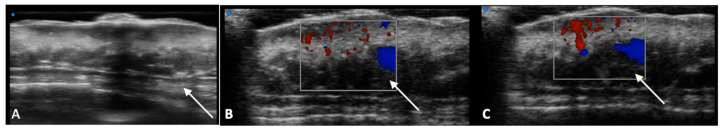
US examination performed with a very high frequency probe (48 Mhz) (**A**) Upper frontal skin site with evident post-actinic scar already treated with previous radiotherapy. (**B**) Round hy-452 hypoechoic area of the hypodermis with non-encapsulated blurred margins (**C**) The lesion shows in-453 intralesional micro-vasculature on Color Doppler control.

**Figure 3 diagnostics-13-00793-f003:**
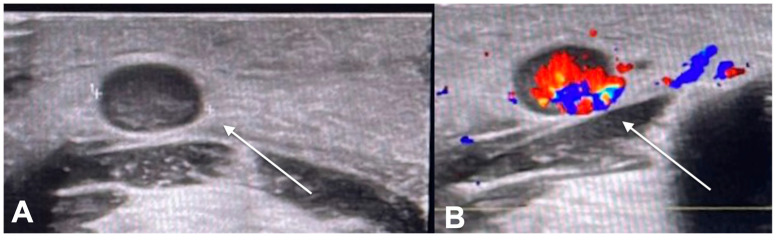
US examination. Submandibular metastatic lymph node of squamous cell carcinoma. (**A**) The lymph node appears round in shape, without hilum differentiation. (**B**) Colorimetric enhancement on Color Doppler examination.

**Figure 4 diagnostics-13-00793-f004:**
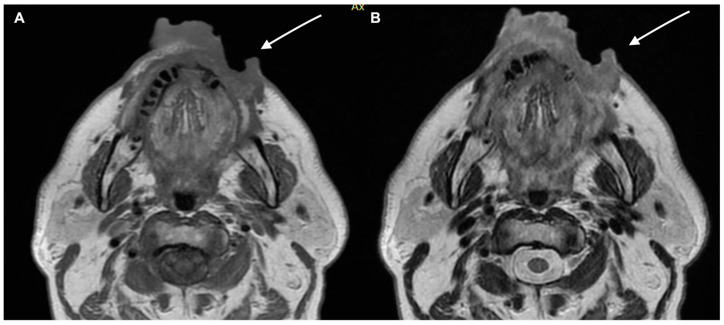
A 65-year-old male patient MRI examination. Lesion interesting the cutaneous and subcutaneous tissues without involving the bone tissue. (**A**) T1 weighted imaging and (**B**) T2 weighted imaging.

**Figure 5 diagnostics-13-00793-f005:**
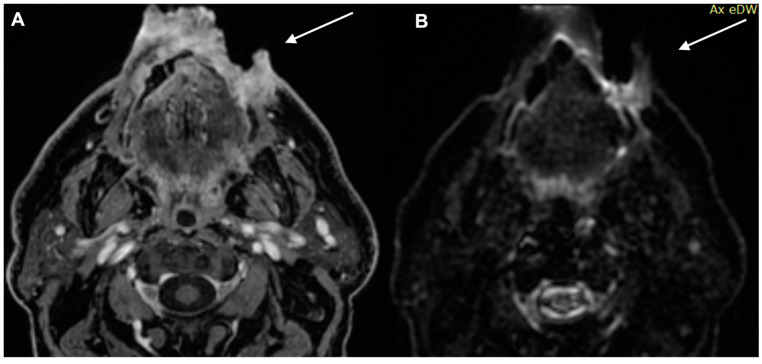
MRI examination of the previous patient. The lesion shows high signal in DWI and inten-sity reinforcement in the contrast-enhanced study image. (**A**) Enhanced MRI imaging and (**B**) DWI.

**Figure 6 diagnostics-13-00793-f006:**
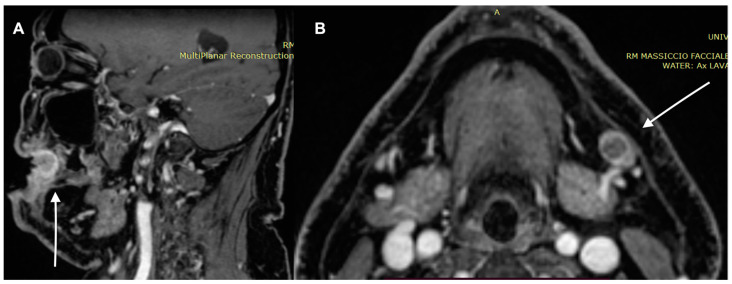
MRI examination of the previous patient. (**A**) Enhanced MRI imaging: sagittal section. The lesion shows high intensity reinforcement in the contrastographic study image. (**B**) Axial section lymph node. Metastatic lymph node with high intensity reinforcement in the contrastographic study image.

**Figure 7 diagnostics-13-00793-f007:**
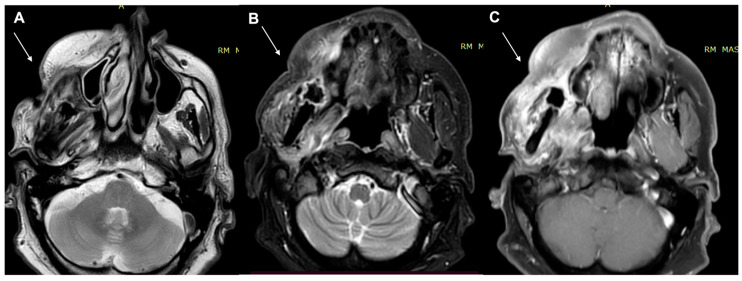
A 53-year-old female patient, MRI examination. (**A**) Enhanced MRI imaging: T2 weighted imaging. (**B**) T2 weighted imaging fat sat. (**C**) Enhanced MRI imaging. The lesion shows high intensity reinforcement in the contrast-enhanced study image (**C**) and bone involvement.

**Figure 8 diagnostics-13-00793-f008:**
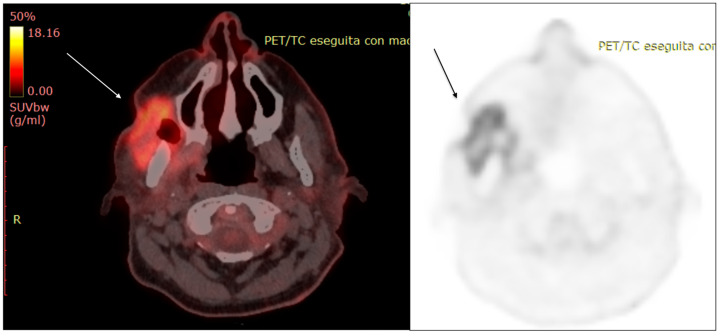
A 53-year-old female patient, PET/TC examination. The exam confirms a cutaneous and subcutaneous lesion with bone involvement.

**Figure 9 diagnostics-13-00793-f009:**
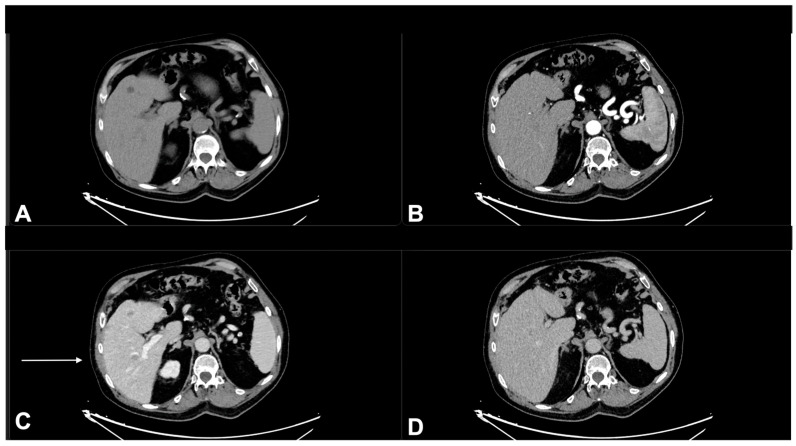
A 78-year-old male patient in treatment with immunotherapy, enhanced CT imaging (128 slices) performed in July 2021. Pre contrast phase CT (**A**), Arterial phase (**B**), Portal phase (**C**), Tardive phase (**D**). In the portal phase (**C**), vague hypodense areas not visible in the other phases of the study, fifth hepatic segment. These areas were suspected of liver metastases from squamous cell carcinoma.

**Figure 10 diagnostics-13-00793-f010:**
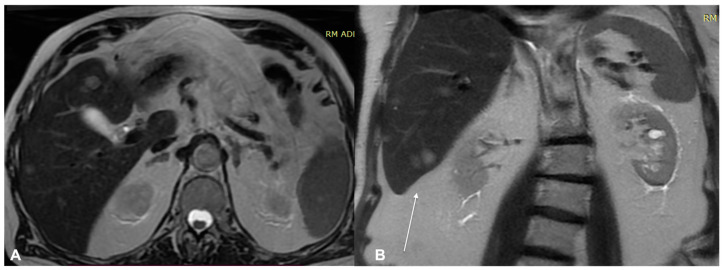
MRI examination of the previous patient performed in August 2021—T2 weighted imaging. Axial section (**A**), Coronal section (**B**). Hypointense areas in the fifth hepatic segment.

**Figure 11 diagnostics-13-00793-f011:**
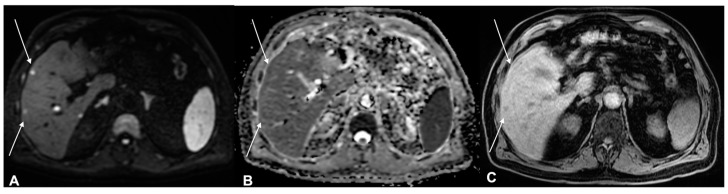
MRI examination of the previous patient performed in August 2021. DWI (**A**), ADC (**B**), enhanced MRI imaging (**C**). The lesions show high signal in DWI, low signal in ADC, and intensity reinforcement in the contrast-enhanced study image. Suspicious areas are confirmed as metastatic lesions.

**Figure 12 diagnostics-13-00793-f012:**
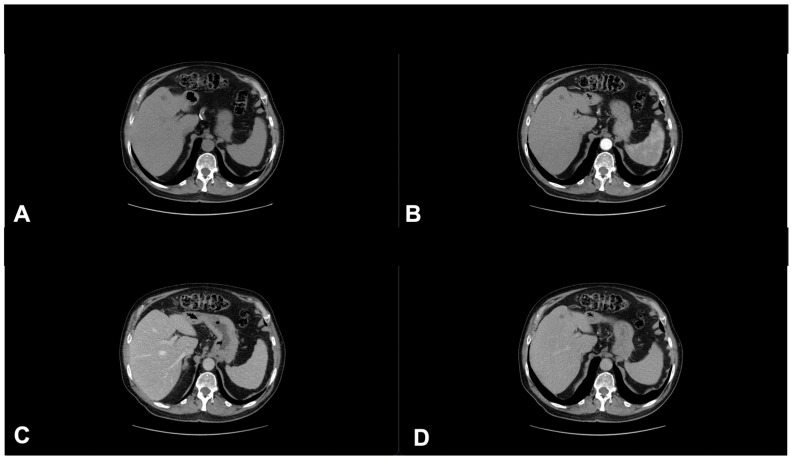
Enhanced CT imaging (128 slices) of the previous patient performed in March 2022. Non enhanced CT (**A**), Arterial phase (**B**), Portal phase (**C**), Tardive phase (**D**). The metastatic lesions were no longer present, pseudo progression.

**Table 1 diagnostics-13-00793-t001:** Characteristics of pseudo progression and hyper progression.

Characteristics of Pseudo Progression and Hyper Progression	
Pseudo progression	Hyper-progression
Pseudo progression is an initial progression in which the tumor burden or the number of tumor lesions increase initially and then decreases over time.	Hyperprogression is a tumor response in which the existing underlying tumor grows rapidly after initiating treatment with an immune checkpoint inhibitor.
Pseudo progression is not true tumour progression, which has been proven by histopathological biopsies that found infiltration and recruitment of various immune cells, such as T or B lymphocytes, in the tumor.	Tumor samples of people who experienced hyperprogression were found to have a greater number of tumor-associated macrophages (macrophages are cells that are part of the immune system that are present in the area surrounding tumors or “tumor microenvironment”).
The occurrence of pseudoprogression has led to the development of immune-related response-evaluation criteria. In this phenomenon, patients treated with immunotherapy experience an initial increase in tumor burden through enlargement of target lesions and/or development of new lesions, followed by a subsequent decrease in the tumor burden qualifying as a partial or complete response.	Hyper progression involves not only the more rapid growth of a tumor but a lower survival rate. In patients developing hyperprogression, immunotherapy treatment should be stopped and the patient should be managed appropriately.

**Table 2 diagnostics-13-00793-t002:** Table of definitions.

Table of Definitions	
iCR	Immune control response is the disappearance of all lesions, measured or unmeasured, and no new lesions.
iSD	Immune stable disease is referred as cancer that is neither decreasing nor increasing in extent or severity.
iPr	unconfirmed progression disease
iUPD	increase of non-target lesions or appearance of new lesion called iUPD
iCPD	Development of another new lesion, increased size of the target or non-target lesions, and/or unequivocal progression of existing non-target lesions.

**Table 3 diagnostics-13-00793-t003:** Immune-related adverse events—irAEs.

irAEs		
Favourable	Neutral	Unfavourable
Developing an irAE	Pruritus	High grade irAEs
Certain irAEs: skin(vitiligo), endocrine, hepatic, gut hypophysitis, and colitis	Taking short term steroids for irAEs	Pre-existing autoimmune disease, i.e., earlier and high prevalence of irAEs
Pre-existing psoriasis	Radiotherapy after treatment	Steroid use for cancer related symptoms
Steroid-sparing therapy		Mucosal melanoma
Combined with radiotherapy		Low PD L1 expression

## Data Availability

The data are available upon request.

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
