# Peer review of "Management of Non-Melanoma Skin Cancer: Radiologists Challenging and Risk Assessment"

_diagnostics, 2023, doi:10.3390/diagnostics13040793_

Round 1

Reviewer 1 Report

Comments:

This is a review paper working on critical analysis of the radiologist’s role in non-melanoma skin cancer patients related to the different patient risk and disease phases. The scientific importance degree on the work is acceptable, while there are proper connectivity between different parts of the text body. The references used in this text have almost high impacts and updated, and overally this work can be considered for publishing at Diagnostics Journal while it meets the standards of this journal.

At abstract section, the last sentences can be expanded focusing on the reviewed issues taking into account the clinical requirements.

Please add more texts about “risk assessment” concept generally and in detail approached in this work.

For better visualization, please indicate lesions at figures, shown in this work.

Author Response

Dear Reviewer, we are happy that you have provided us with suggestions to improve the work; we are truly honoured.
We have changed some sentences to try to be clearer in the exposition and added tables of definitions to make reading easier and more usable; furthermore, as you suggested, we have inserted arrows in the images to better focus on the lesions.

Reviewer 2 Report

# Overall Comments

Thank you for the opportunity to review this manuscript. The authors reported the management of non-melanoma skin cancer from the viewpoint of radiologists. I think that this report is of value, but requires some revisions.

#Major points

As for chapters 4 and 5, it would be better to make the descriptions of the word definitions simplified (for example, using Tables) and the characteristics of hyper-progression, pseudo-progression, and so on are required to be described more.

#Minor points

Page 2, Line 1: sung -> Sung

In the manuscript, the abbreviation should be defined on its first use. And there are quite many misuses of capitalized/non-capitalized words and some unnecessary spacings. Please re-check the manuscript.

Author Response

Dear Reviewer, we are happy that you have provided us with suggestions to improve the work; we are truly honoured.
As you suggested we have changed some sentences to try to be clearer in the exposition and added tables of definitions to make reading easier and more usable; furthermore we have inserted arrows in the images to better focus on the lesions.